# In Vitro Antibacterial Activity Evaluation and Mechanism of Morphology-Controlled Synthesis of Cerium Dioxide Nanoparticles

**DOI:** 10.3390/ijms262110587

**Published:** 2025-10-30

**Authors:** Ziting Gao, Mi Liu, Zhen Wang, Meina Zhang, Genli Shen, Yan Gong, Kaijie Zheng, Min Yang, Qi Wang

**Affiliations:** 1School of Chemistry and Biological Engineering, University of Science and Technology Beijing, Beijing 100083, China; gaoziting2024@163.com (Z.G.); z862955746@163.com (K.Z.); 2National Center for Nanoscience and Technology, Beijing 100190, China; liumi@nanoctr.cn (M.L.); wangzh@nanoctr.cn (Z.W.); zhangmn2024@nanoctr.cn (M.Z.); shengl@nanoctr.cn (G.S.); gongyan2019@nanoctr.cn (Y.G.)

**Keywords:** CeO_2_, morphology, oxygen vacancy, reactive oxygen species (ROS), antibacterial activity

## Abstract

Cerium dioxide (CeO_2_) nanoparticles with distinct morphologies, including rods, cubes, and octahedrons, were synthesized via a straightforward hydrothermal method. The microstructure and morphology of the as-prepared samples were systematically characterized. The antibacterial activity of the samples against *Escherichia coli* was evaluated using the plate counting method. The antibacterial experiments revealed that the antibacterial properties of the samples were arranged in the following order: rod > cube > octahedron. Data analysis indicated that the superior antibacterial performance of the CeO_2_ nanorod was attributed to the higher concentration of oxygen vacancies and adsorption of reactive oxygen species (ROS) on the surface, with ROS playing a critical role in the antibacterial mechanism of CeO_2_. Additionally, density functional theory (DFT) calculations were employed to simulate the oxygen vacancy environments of CeO_2_ with different morphologies and provided indirect insights into ROS behavior. Combining experimental and computational results, a mechanistic framework was proposed to elucidate the dependence relationship between morphology and antibacterial activity of CeO_2_.

## 1. Introduction

Bacterial infection refers to the disease caused by bacteria invading, parasitizing, and reproducing in the human body. It poses a serious threat to human health. Bacteria can enter the human body through different ways, including skin wounds, respiratory tract, digestive tract, etc. Many bacteria release toxins, which can cause direct damage to human tissues or trigger immune responses during growth. For instance, *Escherichia coli*, one of the most common Gram-negative bacteria, is frequently associated with gastrointestinal infections and other illnesses. The discovery of the first organic antibiotic―penicillin [1]—revolutionized medicine and alleviated the suffering of billions worldwide. However, the subsequent emergence of antibiotic resistance has become an inevitable and pressing challenge. After extensive research, scientists have identified metal and metal oxide nanomaterials as promising candidates for future antibiotic alternatives due to their unique mechanisms of action, high-temperature resistance, broad-spectrum antibacterial properties, and low propensity for inducing drug resistance [2]. Nanomaterials such as gold, silver, iron, MnO [3], TiO_2_ [4], and ZnO [5] have been well used in clean environments and biomedical applications. Despite excellent antibacterial properties, some materials still exhibit toxicity to normal cells at low doses. For example, silver nanoparticles (AgNPs) are traditional antibacterial agents which involve various antibacterial mechanisms and can effectively treat or prevent infection; however, the biological toxicity of AgNPs has been included in the investigation criteria. The National Institute for Occupational Safety and Health (NIOSH) evaluated over 100 studies on silver nanomaterials in animal cells and concluded that these materials pose potential health risks [6]. Therefore, the urgent need for new and safe antibacterial agents is increasing.

Rare earth elements have been applied in the field of antibacterial materials and have achieved good results [7]. Ceria (CeO_2_) is a member of the lanthanide metal oxides, which shows the potential of redox behavior and good applications in biomedical fields such as anti-cancer, anti-inflammatory, anti-oxidation, antibacterial, anti-diabetes, and treatment of various neurodegenerative diseases [8,9].

The reversible transition between the reduction (Ce^3+^) and oxidation (Ce^4+^) states of cerium ions in CeO_2_ nanoparticles endows them with effective active oxygen scavenging and anti-oxidative stress cell protection [10,11]. Pang et al. directly observed the formation and migration of oxygen vacancies on the surface of CeO_2_ using in situ environmental transmission electron microscopy (ETEM). They found that oxygen vacancies not only provide active sites but also enhance the chemical reactivity of the material by altering its surface electronic structure [12]. Qin et al. effectively converted glucose into hydroxyl radicals by loading glucose oxidase in mesoporous CeO_2_ hollow sphere nanozymes. The resulting hydroxyl radicals would seriously damage the cell structure of bacteria and hinder the formation of biofilm, which reflect the important role of reactive oxygen species in the antibacterial process [13]. Madhura A. Damele et al. used surface modification to adjust the Ce^3+^/Ce^4+^ ratio and found that the main factor of antibacterial activity was mainly related with the presence of oxygen vacancies in nanoparticles [14]. In addition, oxygen vacancies have also been shown to increase catalytic active sites, accelerate electron transfer [15], and reduce the reaction barrier [16]. Although the antibacterial mechanisms involving the oxygen vacancy of CeO_2_ have been more recognized, the influences of oxygen vacancies on the different crystal planes of CeO_2_ in antibacterial behavior still need to be described further.

This study systematically investigates the antibacterial properties of CeO_2_ with different morphologies (octahedral, rod-shaped, and cubic) using the plate coating method, with *Escherichia coli* as the experimental subject. The differences in antibacterial activity and crystal structure characteristics are analyzed in detail. Additionally, density functional theory (DFT) simulations are employed to model the crystal surface structures, and the formation energy of oxygen vacancies on specific exposed crystal planes is calculated. The antibacterial mechanism is further explored from the perspective of oxygen vacancy concentration and distribution. The innovation of this research lies in revealing the intrinsic relationship between crystal plane exposure, oxygen vacancy dispersion, reactive oxygen species generation, and antibacterial efficacy by comparing the antibacterial performance of CeO_2_ with different morphologies. This provides a new theoretical perspective for a deeper understanding of the antibacterial effects of CeO_2_ nanoparticles and lays a significant scientific foundation for the design and development of highly efficient antibacterial materials.

## 2. Results and Discussion

### 2.1. Antibacterial Activity of CeO_2_-Rod, CeO_2_-Cube, and CeO_2_-Oct

The antibacterial agents CeO_2_-Oct, CeO_2_-Rod, and CeO_2_-Cube were diluted by concentration gradient with *Escherichia coli* as the experimental bacteria, and the antibacterial properties were evaluated. The detailed results are shown in Figure 1 and Table 1.

The blank control group, without the addition of any nanomaterials, did not exhibit any antibacterial activity. Figure 1A–C show the antibacterial activity of CeO_2_-Oct. It can be seen that CeO_2_-Oct has almost no antibacterial properties at three different concentrations. When exposed to different concentrations of the CeO_2_-Rod solution, bacterial colonies could not survive even at the lowest concentration (Figure 1D–F), indicating that CeO_2_-Rod exhibited good antibacterial properties. When *E. coli* contacted with three different concentrations of the CeO_2_-Cube antibacterial solution (Figure 1G–I), it can be seen that CeO_2_-Cube only showed certain antibacterial activity—the higher the concentration of the solution, the better the antibacterial performance against *E. coli.* When the concentration of the CeO_2_-Cube antibacterial solution is 32 mg/mL, the antibacterial efficiency reaches 100%. When the concentration is 8 mg/mL, the inhibition rate is almost 0, as shown in Table 1. The minimum inhibitory concentration (MIC) of CeO_2_-Rod is 1/5 of that of CeO_2_-Cube. The results of antibacterial experiments show that the antibacterial properties of the samples were in the following order: CeO_2_-Rod > CeO_2_-Cube > CeO_2_-Oct.

### 2.2. Characterization of CeO_2_ Antibacterial Solutions with Different Morphologies

The crystal structures of CeO_2_-Rod, CeO_2_-Cube, and CeO_2_-Oct were characterized by XRD. As shown in Figure 2, the diffraction peaks at 2θ = 28.5°, 33.0°, 47.4°, 56.4°, 59.2°, 69.5°, 76.6°, and 79.1° in the XRD profile of cerium oxide clearly demonstrate the presence of the cubic fluorite structure of CeO_2_ (PDF # 81-0792).

The lattice parameters of each sample were calculated according to the XRD data, as shown in Table 2. The results show that the lattice parameter values of CeO_2_-Oct, CeO_2_-Rod, and CeO_2_-Cube are 5.4144 Å, 5.4276 Å, and 5.4193 Å, respectively, which are higher than the theoretical value of the CeO_2_ lattice parameter (5.411 Å) [17]. In particular, the lattice parameter value of CeO_2_-Rod increases more than the values of CeO_2_-Oct and CeO_2_-Cube. The ionic radius of Ce^3+^ is larger than that of Ce^4+^, resulting in lattice expansion and an increase in the lattice parameter values. Therefore, the concentration of Ce^3+^ ions is positively correlated with an increase in the lattice parameter values [18]. Additionally, oxygen vacancy is easily formed due to the transformation of Ce^4+^ ions to Ce^3+^ ions in the three kinds of CeO_2_ nanomaterials. The microstrain (ε) values of these samples can be used to indicate the concentration of oxygen vacancy; these values were determined from line-broadening measurements on the different crystal planes by using the equation shown in Table 2 [19]. The lattice strain value of CeO_2_-Rod (ε = 0.182) is much higher than the values of CeO_2_-Oct (ε = 0.177) and CeO_2_-Cube (ε = 0.179), suggesting that the oxygen vacancy density of CeO_2_-Rod is greater than the densities of CeO_2_-Oct and CeO_2_-Cube.

The morphology and structure of the samples were characterized by SEM and TEM. In the SEM image (Figure 3a) and TEM image (Figure 3b) of CeO_2_-Oct, it can be seen that the average particle size of CeO_2_-Oct is about 100 nm. The exposed crystal plane in the HRTEM (Figure 3c) is determined to be the (111) crystal plane according to the interplanar distance 0.312 nm. The SEM and TEM images of CeO_2_-Rod are shown in Figure 3d and Figure 3e, respectively. It can be seen that the average diameter of CeO_2_-Rod is about 20 nm and the length is about 150–300 nm. In the HRTEM of CeO_2_-Rod (Figure 3f), when observed along the (100) direction, the exposed (200) crystal plane can be determined obviously, which is consistent with the previous observation results [20]. Additionally, the fringe lattice spacing at a 45° to the extension direction of the nanorods is 0.19 nm, corresponding to the (220) crystal plane. Figure 3g,h show the SEM and TEM images of CeO_2_-Cube. The size of CeO_2_-Cube is about 50–60 nm with a cubic morphology. In the HRTEM of CeO_2_-Cube (Figure 3i), only the (100) crystal plane can be observed.

In the fluorite-type cubic structure of CeO_2_, there are three low-index crystallographic planes: the highly stable (111) plane, the moderately stable (110) plane, and the less stable (100) plane [20]. It has been reported that the energy required to form oxygen vacancies on the (110) and (100) planes is significantly lower than that on the (111) plane. As a result, oxygen vacancies are more easily formed on the (100) plane, followed by the (110) plane, and finally on the (111) plane [21]. The antibacterial experiment results mentioned previously have demonstrated that CeO_2_-Rod (both the exposed (110) and the (100) planes) and CeO_2_-Cube (exposed (100) plane) exhibit significantly higher antibacterial activity compared with that of CeO_2_-Oct (exposed (111) plane). This enhanced antibacterial performance may be attributed to the higher reactivity of the exposed planes, which facilitates the formation of oxygen vacancies more effectively to improve antibacterial properties. However, the superior antibacterial activity of CeO_2_-Rod over CeO_2_-Cube cannot be solely explained in the view of the crystal plane effect. Surface defects, such as oxygen vacancies and other structural irregularities, may also play a synergistic role in enhancing antibacterial performance. Further insights into the antibacterial mechanism can be obtained through detailed analysis using Raman spectroscopy and X-ray photoelectron spectroscopy (XPS), which can help elucidate the contributions of surface chemistry and defect states to the observed antibacterial behavior.

The Raman scattering method is used to identify the solid solution phase, which can indirectly reflect the properties of oxygen vacancies. Figure 4 shows the visible Raman spectra of all samples. In this pattern, a strong peak at ca. 460 cm^−1^ and a weak peak at ca. 610 cm^−1^ can be distinguished. The peaks at 460 cm^−1^ and 610 cm^−1^ correspond to the fluorite F_2g_ mode and the defect-induced mode (D mode), respectively [22]. For CeO_2_, Raman bands at ca. 610 cm^−1^ are assigned to the presence of oxygen vacancies caused by the conversion of Ce^4+^ to Ce^3+^ [23]. The ratio of the integrated peak area of the oxygen vacancy (~610 cm^−1^) to that of the main peak area (460 cm^−1^) is defined as A_610_/A_460_, which is used to characterize the relative concentration of oxygen vacancies in these samples [19]. The ratio of A_610_/A_460_ is ranked in the order of CeO_2_-Rod > CeO_2_-Cube > CeO_2_-Oct (Table 3), indicating that CeO_2_-Rod exhibits a higher oxygen vacancy concentration, which is consistent with the previously mentioned XRD results.

The oxidation state of the surface components of CeO_2_ was investigated by XPS analysis. Figure 5a is the XPS spectra of Ce3d core layers of all samples. The peaks V_0_/V_0_′, V_1_/V_1_′, and V_2_/V_2_′ refer to three pairs of spin-orbit doublets, which can be attributed to surface Ce^4+^ [24,25], while U_0_/U_0_′ and U_1_/U_1_′ can be ascribed to Ce^3+^ [26]. The relative content of Ce^3+^ in all samples can be calculated according to Equation (1) and is shown in Table 3. The data show that the content of Ce^3+^ in CeO_2_-Rod is higher than the content values of CeO_2_-Oct and CeO_2_-Cube, which is also consistent with the previously mentioned discussion results. The relative content of Ce^3+^ in CeO_2_ is related to oxygen vacancies because the formation of Ce^3+^ is usually accompanied by the presence of oxygen vacancies due to a balancing of the charge, which affects the concentration of Ce^3+^. Therefore, the higher the content of Ce^3+^, the more oxygen vacancies the samples possess.(1)XCe3+=ACe3+SCe∑A(Ce3++Ce4+)SCe×100%
where $X_{{Ce}^{3+}}$ is the percentage content of Ce^3+^, A is the integrated area of the characteristic peak in the XPS pattern, and S is the sensitivity factor (S = 7.399) [19].

Figure 5b displays the O1s XPS spectra of the CeO_2_-Oct, CeO_2_-Rod, and CeO_2_-Cube samples. In all samples, the peak at the binding energy in the range of 529.0–530.0 eV corresponds to lattice oxygen (O_α_), which represents oxygen atoms in the CeO_2_ crystal lattice. The peak at the binding energy in the range of 531.0–532.0 eV corresponds to surface-adsorbed oxygen (O_β_), which is associated with oxygen species adsorbed on surface oxygen vacancies [26]. The concentration of adsorbed oxygen is estimated by calculating the integral ratio of (O_β_/(O_α_ + O_β_)) (Table 3). The results show that the CeO_2_-Rod sample contains more surface-adsorbed oxygen species compared with CeO_2_-Cube and CeO_2_-Oct and indicate that more oxygen vacancies exist on the surface of CeO_2_-Rod. These oxygen vacancies as active sites, facilitate the adsorption and activation of oxygen molecules to generate reactive oxygen species (ROS, such as ·OH and O_2_^−^), which play a critical role in antibacterial mechanisms by oxidizing lipids and proteins of bacterial cell membranes, disrupting cellular structures, and inhibiting biofilm formation [27]. The O1s XPS spectra can further support the pivotal role of surface defects, especially oxygen vacancies, in enhancing the antibacterial properties of CeO_2_.

Electron paramagnetic resonance (EPR) spectroscopy, known for its high sensitivity to unpaired electrons, has been widely used for the detection of paramagnetic reactive oxygen species (ROS). In this study, EPR was also employed to analyze the types of ROS of CeO_2_ nanomaterials with three different morphologies (CeO_2_-Rod, CeO_2_-Cube, and CeO_2_-Oct), including superoxide anion radicals (·O_2_^−^), singlet oxygen (^1^O_2_), and hydroxyl radicals (·OH) (Figure 6a–c). In the ·O_2_^−^ spectra (Figure 6a), the signal intensities of the samples follow this order: CeO_2_-Rod > CeO_2_-Cube > CeO_2_-Oct. This difference is attributed to the highly active crystal planes (such as (110) and (100)) and abundant oxygen vacancies. Oxygen vacancies as active sites, can promote the adsorption of O_2_ and facilitate the single-electron reduction reaction (O_2_ + e^−^→·O_2_^−^). In contrast, CeO_2_-Oct, dominated by the (111) crystal plane, exhibits a lower concentration of oxygen vacancies, resulting in less ·O_2_^−^ generation capability. The singlet oxygen (^1^O_2_) signal intensity shows no significant difference among the three CeO_2_ nanomaterials (Figure 6b). The generation of ^1^O_2_ primarily relies on energy transfer mechanisms (oxidation of H_2_O/OH^−^ by photo generated holes) [28]. Since the light absorption and carrier separation efficiency of CeO_2_ with different morphologies are similar, the signal intensities are also similar. The characteristic peak intensity of hydroxyl radicals (·OH) of CeO_2_-Rod is higher than the intensity values of CeO_2_-Cube and CeO_2_-Oct (Figure 6c). The generation of ·OH is closely related to the oxygen vacancy concentration in CeO_2_. The high oxygen vacancy concentration in CeO_2_-Rod not only enhances its superoxide dismutase (SOD)-like activity, enabling the catalytic conversion of superoxide anions (O_2_^−^) into hydrogen peroxide (H_2_O_2_) via a dismutation reaction, but also promotes the activation and decomposition of the in situ generated H_2_O_2_ into highly reactive hydroxyl radicals (·OH) through a Fenton-like reaction [29,30]. Through the above analysis, it is evident that CeO_2_-Rod, due to its high oxygen vacancy concentration and exposure of active crystal planes, exhibits significantly superior ROS generation capability compared with CeO_2_-Cube and CeO_2_-Oct. These findings are consistent with our previous analysis.

### 2.3. DFT Calculations

The difference charge density shows that there is an obvious charge transfer in the process of forming O vacancies on different crystal planes, that is, the electrons on the oxygen atom are transferred to the two adjacent cerium ions, resulting in the formation of oxygen vacancies and a decrease in the oxidation state of cerium ions (Figure 7).

An O atom was removed from different surfaces of CeO_2_ to form O vacancies. The difficulty of vacancy defect formation was evaluated by defect formation energy, and the formation energy of different vacancies was calculated (Figure 8). The formation energies of oxygen vacancies on the (100), (110), and (111) crystal planes are 3.94 eV, 2.62 eV, and 4.73 eV, respectively. That is to say, the order of the formation energy of oxygen vacancies (E_f_) on different exposed crystal planes is: (110) < (100) < (111). This indicates that oxygen vacancies are more easily formed on the (110) crystal plane, which is related to the surface energy and atomic arrangement of the crystal plane. Therefore, CeO_2_-Rod [exposed (110) and (100) facets] is expected to exhibit better antibacterial properties than CeO_2_-Cube [exposed (100) facets] and CeO_2_-Oct [exposed (111) facets], which is consistent with the antibacterial results.

In the EPR test, we found that the intensity of the ·OH characteristic tetragonal peak corresponding to CeO_2_-Rod is much higher than that of CeO_2_-Oct and CeO_2_-Cube, so the adsorption energy of hydroxyl groups on different crystal plane vacancies of CeO_2_ was calculated by simulation (Figure 9). According to the theoretical calculation, the hydroxyl adsorption energies of CeO_2_ on (100), (110) and (111) crystal planes are −2.59 eV, −4.31 eV and −1.64 eV, respectively. These values indicate that the adsorption of hydroxyl groups on the (110) crystal plane is the most stable. This may be related to the surface structure and electronic properties of the (110) crystal plane, which enables it to form chemical bonds with hydroxyl groups more effectively.

Therefore, CeO_2_-Rod (exposed (110) and (100) crystal planes) exhibits higher hydroxyl adsorption energy compared with CeO_2_-Cube (exposed (100) crystal plane), and CeO_2_-Oct (exposed (111) crystal plane). This enhanced hydroxyl adsorption facilitates the reaction between hydroxyl groups and organic components in bacterial cell walls, leading to the disruption of cellular structures and subsequent bacterial inactivation. Consequently, CeO_2_-Rod demonstrates superior antibacterial performance, which is consistent with the aforementioned characterization and antibacterial results.

### 2.4. Antibacterial Mechanism Analysis

Under the same conditions, the three morphologies of CeO_2_ exhibit different antibacterial properties. In a word, reactive oxygen species (ROS) play a critical role in the antibacterial process (Figure 10). Firstly, the conversion between Ce (III) and Ce (IV) in CeO_2_ can produce oxygen vacancy, on the surface of which a gas oxygen molecule can be activated into ROS. These ROS contact with bacteria, penetrate bacterial cells, disrupt their cellular structures and physiological functions, and finally cause bacterial death [31,32,33,34]. Additionally, CeO_2_ interacts with negatively charged groups on bacterial surfaces (such as phosphate and carboxyl groups) through electrostatic forcing to destroy the integrity of the cell membrane [35,36]. The differences in antibacterial behavior and corresponding mechanisms of CeO_2_ with different morphologies are discussed in detail.

Oxygen vacancy is a common defect type in metal oxides and plays an important role in the generation of reactive oxygen species (ROS) [37]. Firstly, the formation of oxygen vacancies is shown in Figure 11. When the oxygen atom leaves its lattice position, two electrons are left behind, which are then distributed by two Ce^4+^ ions to form Ce^3+^. Therefore, the number of oxygen vacancies is closely related to the concentration of Ce^3+^. The presence of oxygen vacancies breaks the charge balance of the lattice, increases the local electron density, exhibits a negative charge in a small range, reduces the activation energy of the reaction, accelerates the generation of ROS, and enhances the activation ability of the reactants [27]. The data obtained from XRD, the Raman test, and XPS show that the concentrations of oxygen vacancy and Ce^3+^ are the highest in CeO_2_-Rod with the best antibacterial property. In other words, more ROS are easily formed on the surface of CeO_2_-Rod and participate in antibacterial reactions. Therefore, the antibacterial mechanism is described further in the view of ROS.

The corresponding mechanism is described as follows: CeO_2_ nanoparticles, owing to their unique electronic structure, exhibit two variable valence states, Ce^3+^ and Ce^4+^, which contribute to their remarkable catalytic activity. Specifically, during the valence transition between Ce^3+^ and Ce^4+^, CeO_2_ nanoparticles can effectively catalyze the conversion of an oxygen molecule (O_2_) into a superoxide anion (O_2_^·−^), as shown in Equation (2) [38]. When *Escherichia coli* grows, it can metabolize glucose to produce acidic substances (such as acetic acid and lactic acid) and lead to a decrease in pH of the culture medium. As the pH drops, the concentration of H^+^ is increased around the bacteria, which promotes the conversion of the superoxide anion (O_2_^·−^) into hydrogen peroxide (H_2_O_2_), as described in Equation (3). Furthermore, the generated hydrogen peroxide, under the catalytic effect of Ce^3+^, undergoes a Fenton-like reaction and produces hydroxyl radicals (·OH). These hydroxyl radicals are highly active oxidizing agents, which can effectively kill bacteria, as illustrated in Equation (4) [39].(2)$${Ce}^{3+}+O_{2}\to {Ce}^{4+}+{O_{2}}^{\cdot -}$$(3)$${O_{2}}^{\cdot -}+2H^{+}\to H_{2}O_{2}$$(4)$$H_{2}O_{2}+{Ce}^{3+}\to {Ce}^{4+}+\cdot OH+{OH}^{-}$$

The resulting strong oxidizing ROS directly act on the bacterial cell wall and cell membrane, lead to the tearing and destruction of the structure, and exert antibacterial effects [40]. Under normal physiological conditions, the production and removal of ROS in bacteria are in a dynamic balance. However, excessive ROS will be produced due to the addition of CeO_2_, which can lead to oxidative stress and cause damage to bacteria. The hydroxyl radical (·OH) reacts with lipids and proteins on the surface of the cell membrane or near the cell membrane. As a form of oxygen species, singlet oxygen (^1^O_2_) can freely diffuse through the cell membrane. Once it enters the cell, ^1^O_2_ can react with unsaturated fatty acids, proteins, and other intracellular molecules and cause oxidative damage. Superoxide free radicals (O_2_^•−^) are negatively charged and cannot directly penetrate the phospholipid bilayer of the cell membrane, but they can enter the cell through specific channels or carrier proteins on the cell membrane. Upon entering the cell, O_2_^•−^ can react with other molecules and convert into other ROS, such as hydrogen peroxide (H_2_O_2_), which may produce hydroxyl radicals to damage the bacteria further.

## 3. Materials and Methods

### 3.1. Materials

Cerium nitrate hexahydrate (99.5%, AR), sodium hydroxide (96%, AR), trisodium phosphate dodecahydrate (98%, AR), and anhydrous ethanol (AR) were produced by Shanghai Macklin Biochemical Technology Co., Ltd., Shanghai, China. The reagents used in this experiment are all AR grade. Luria-Bertani (LB) agar and nutrient agar broth were purchased from Sinopharm Group Chemical Reagents Co., Ltd., Beijing, China, and *Escherichia coli* DH5α (MTCC 1652) from the China Microbiology Center, Beijing, China. All solutions were prepared with deionized water.

### 3.2. Synthesis of CeO_2_ with Rod and Cube Morphology (CeO_2_-Rod and CeO_2_-Cube)

The CeO_2_ samples were synthesized by a hydrothermal process. Briefly, a mixture of 1.736 g cerium nitrate hexahydrate, 16.8 g sodium hydroxide, and 100 mL distilled water was added to a Teflon stainless steel autoclave and sealed. The resulting mixture was heated to 120 °C for CeO_2_-Rod and 160 °C for CeO_2_-Cube and then kept at the given temperature for 24 h. The precipitates in the autoclave were collected, washed several times with anhydrous ethanol and deionized water, and then dried in air at 80 °C for 12 h. The preparation process is shown in Figure 1.

### 3.3. Synthesis of CeO_2_ with Octahedral Morphology (CeO_2_-Oct)

A quantity of 0.868 g (2 mmol) cerium nitrate hexahydrate as well as 0.0076 g (0.02 mmol) sodium phosphate (used as precipitant) were dissolved in 100 mL deionized water and stirred at room temperature for 0.5 h. The solution was then poured into the reactor for hydrothermal reaction (170 °C, 12 h). The precipitates in the autoclave were collected, washed with anhydrous ethanol and distilled water, and then dried at 80 °C for 12 h to obtain the sample.

### 3.4. Characterization

The crystal structures of the CeO_2_ samples were characterized using powder X-ray diffraction (XRD, Philips X’pert PRO equipped with Cu Kα radiation (λ = 1.5406 Å), PANalytical B.V. Almelo Netherlands). The XRD patterns were recorded in the 2θ range of 20° to 80° at a scanning speed of 0.17°/s. Morphological analysis was performed using scanning electron microscopy (SEM, Merlin, Carl Zeiss AG, Oberkochen, Germany) and transmission electron microscopy (TEM, HT7700, Hitachi High-Tech Corporation, Tokyo, Japan). High-resolution transmission electron microscopy (HRTEM, T-20, America Philips Corporation, Hillsboro, OR, USA) was employed to analyze the crystal planes of the samples. The elemental concentrations of the samples were quantified using inductively coupled plasma mass spectrometry (ICP-MS, NexION 300X, PerkinElmer Inc., Waltham, MA, USA). The surface composition and chemical states of the samples were determined by X-ray photoelectron spectroscopy (XPS, ESCALAB250Xi, Thermo Fisher Scientific, Waltham, MA, USA), with the binding energy calibrated using the C1s peak at 284.8 eV as a reference. Raman spectra were acquired using a Raman spectrometer (T64000, HORIBA Ltd., Kyoto, Japan) to analyze the relative content of Ce^3+^ and Ce^4+^. Reactive oxygen species (ROS) were identified using electron paramagnetic resonance (EPR) spectroscopy (EPR200M, CIQTEK Co., Ltd., Hefei, China) at room temperature, with measurements performed at a frequency of 9.8 GHz and a magnetic field modulation of 100 kHz.

### 3.5. Details of DFT + U Calculation

All the density functional theory calculations were performed using the Vienna ab initio Simulation Package (VASP) [41]. The projector augmented-wave (PAW) method with a plane-wave basis set was used [42,43], utilizing a kinetic energy cutoff of 500 eV for the basis set construction. The Perdew–Burke–Ernzerhof (PBE) [44] exchange–correlation functional with the generalized gradient approach (GGA) was applied to describe the electronic structure. A Γ centered k-points to integrate the Brillouin zone. The energy and force convergence criteria were set as 1.0 × 10^−5^ eV and 0.02 eV Å^−1^, respectively. We used a large vacuum gap of 15 Å to eliminate the interactions between neighboring slabs. Because cerium has f electrons, it could strongly affect the electron distribution of the partially reduced CeO_2_ surface. Therefore, a Hubbard U term correction using the formalism of Dudarev et al. was also used [45]. The value of U = 5.0 eV was used in this work according to the previously tested U values.

The defect formation energy $\left(E_{f}\right)$ of oxygen vacancy is defined as Equation (5):(5)$$E_{f}=E_{s\_V_{O}}+\frac{1}{2}E_{O_{2}}-E_{S}$$
where $E_{s\_V_{O}}$ and $E_{S}$ are the total energies of the supercell with and without an oxygen vacancy and $E_{O_{2}}$ is the total energy of an O_2_ molecule.

The adsorption energy (E_ads_) of a molecule on the ceria surface is defined as Equation (6):
(6)$$E_{ads}=E_{mol@ceria}-(E_{ceria}+E_{mol})$$
where $E_{ceria}$, $E_{mol}$, and $E_{mol@ceria}$ are the total energies for the isolated ceria slab, isolated molecule, and ceria with the adsorbed molecule, respectively.

### 3.6. Antibacterial Test

The antibacterial activity of CeO_2_ nanoparticles with different morphologies on *Escherichia coli* (*E. coli*, Gram-negative bacteria) was evaluated through the coating plate method [46]. All materials were autoclaved at 0.1 MPa at 121 °C for 60 min before use. The bacterial suspension without any antibacterial agent was used as the blank group, and the bacterial suspension with the antibacterial agent was used as the experimental group. Firstly, *E. coli* was obtained from a microbial detection center and activated. It was then transferred to a nutrient broth medium to prepare the bacterial suspension, which was diluted with phosphate-buffered saline (PBS) to a concentration of 1 × 10^5^ CFU/mL. A 100 μL aliquot of the prepared bacterial suspension was mixed with 0.90 mL of the diluted sample suspension, and the mixture was subjected to shaking culture. After a specific period of contact, 100 μL of the bacterial and antibacterial solution mixture was spread onto Luria-Bertani (LB) agar plates. Following incubation at 37 °C for 24 h, viable colonies on the LB plates were observed and counted. The experiments were conducted in triplicate to ensure the reliability of the antibacterial results.

## 4. Conclusions

In this study, the antibacterial activities of CeO_2_ nanostructures with different morphologies (CeO_2_-Oct, CeO_2_-Rod, and CeO_2_-Cube) were systematically evaluated and compared. Various experimental characterization techniques were employed to analyze their microstructures and surface chemical properties in detail. Density functional theory (DFT) was used to calculate and simulate the distribution of oxygen vacancies and electronic environments in CeO_2_. The antibacterial performance against *Escherichia coli* (*E. coli*) was assessed using the plate coating method. The results demonstrated that CeO_2_-Rod exhibited significantly higher bactericidal activity compared with CeO_2_-Cube and CeO_2_-Oct. This difference was attributed to the exposure of highly active crystal facets (such as (110) and (100)) and a higher concentration of oxygen vacancies in CeO_2_-Rod. The generation of ROS, induced by the high concentration of oxygen vacancies, is the core factor influencing the antibacterial behaviors of CeO_2_. Specifically, (·O_2_^−^) and (·OH) disrupt bacterial cell walls and membrane structures through oxidative stress reactions, lead to lipid peroxidation, protein denaturation, and DNA damage, ultimately resulting in cell death.

In summary, the generation of reactive oxygen species (ROS) induced by high oxygen vacancy concentrations plays a crucial role in the antibacterial mechanism of CeO_2_. Among the ROS, superoxide anion radicals (·O_2_^−^) and hydroxyl radicals (·OH) are particularly critical, as they cause oxidative stress that damages bacterial cell walls and membranes, ultimately leading to cell death.

## Data Availability

The original contributions presented in this study are included in the article.

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
