# Peer review of "In Vitro Antibacterial Activity Evaluation and Mechanism of Morphology-Controlled Synthesis of Cerium Dioxide Nanoparticles"

_ijms, 2025, doi:10.3390/ijms262110587_

Round 1
Reviewer 1 Report
Comments and Suggestions for Authors
Studies on the antimicrobial properties of cerium oxide nanoparticles show great promise. The authors have conducted extensive research in this area, which merits publication in the International Journal of Molecular Sciences (IJMS). However, the manuscript requires major revisions before publication.
The presented data raises an important question. Considering the physical and chemical processes involved in synthesizing cerium oxide nanoparticles, the reason for the lowest concentration of trivalent cerium ions in the CeO2-Oct sample remains unclear (see Table 3). This sample's precursors were loaded in an autoclave in a nearly neutral medium. Moreover, after the decomposition of cerous nitrate, the environment became acidic due to the release of nitric acid, hindering the oxidation of cerous ions. Meanwhile, the other two samples were obtained under harsh conditions in the presence of approximately 4 M NaOH, where cerous ion oxidation occurs rapidly and completely. Is there a typo or confusion here? Were some additional synthesis conditions omitted from the description? Clarification is required.
In the introduction, the authors should revise the literature review to ensure its relevance. For example, the statement "Rare earth elements have been applied in the field of antibacterial materials and achieved good results" is referenced [7]. However, this reference is to a study on the use of rare-earth–platinum alloy nanoparticles in mesoporous zeolite for the catalytic propane dehydrogenation reaction.
The same remark applies to the Results and Discussion section. For instance, the text "Therefore, the concentration of Ce3+ ions is positively correlated with the increase of lattice parameters" is accompanied by a reference to research on the green synthesis of CeO2 NPs and SiO2 NPs from leaf extracts of Nyctanthes arbortristis [24].
It is also important to note that size-dependent lattice expansion is observed in many ionic compound nanoparticles. Back in 2012, Diehm et al. showed that surface stress due to negative Madelung pressure is the main cause of lattice expansion [https://doi.org/10.1002/cphc.201200257]. They also excluded point defects of various cerium charge states as a general explanation. One more comment on the mechanisms: On line 402, the authors state: " CeO2 can bind with sulfhydryl groups on proteins and further intensify the leakage of cellular contents". Unlike silver, however, cerium has no affinity for sulfhydryl groups. Its thiophilicity is 0.1, which is too low [see Table S4 in https://doi.org/10.1021/acs.inorgchem.6b01702]. Future studies should strongly consider using modern knowledge on the behavior of cerium oxide nanoparticles, including the mechanisms of antioxidant and antimicrobial activity.
On page 6, line 211, the authors state that the lattice parameter of CeO2-Rod increases more than those of CeO2-Rod and CeO2-Cube. However, the value cannot exceed itself, therefore, the typo should be corrected.
There are duplicate reference numbers in columns and in parentheses in the References. The excess numbers need to be removed.
Bacterial species names, such as Escherichia coli, should be italicized in the text.
The final sentence of the conclusion reads: "The CeO2 nanostructures reported in this study… hold promise for diverse applications, including drug delivery, anticancer and anti-inflammatory therapies, antifungal activity, optoelectronics, gas sensing, and cosmetics". However, since these claims are not supported by experiments in the submitted manuscript, they should be removed or relocated to other sections as speculation or discussion.
The text needs to be checked for grammar errors. There are numerous missing spaces, such as consecutive spellings of words (e.g., line 318: "CeO2-Cubeand"), errors in superscripts and subscripts, and others. Proofreading by a native speaker is also recommended.
Comments on the Quality of English LanguageProofreading by a native speaker is recommended.
Author Response
Thank you very much for taking the time to review this manuscript. Please
find the detailed responses below and the corresponding
revisions/corrections highlighted/in track changes in the re-submitted files.
Comments 1: [The presented data raises an important question. Considering the physical and
chemical processes involved in synthesizing cerium oxide nanoparticles, the reason for the lowest
concentration of trivalent cerium ions in the CeO2-Oct sample remains unclear (see Table 3). This
sample's precursors were loaded in an autoclave in a nearly neutral medium. Moreover, after the
decomposition of cerous nitrate, the environment became acidic due to the release of nitric acid,
hindering the oxidation of cerous ions. Meanwhile, the other two samples were obtained under
harsh conditions in the presence of approximately 4 M NaOH, where cerous ion oxidation occurs
rapidly and completely. Is there a typo or confusion here? Were some additional synthesis
conditions omitted from the description? Clarification is required.]
Response 1: [Before the writing of this paper, after a lot of experiments and
comparison of different synthesis methods, the hydrothermal method was finally
selected to synthesize nanoparticles with good dispersibility. Different from the
traditional precipitation plus high-temperature calcination ( above 500 °C ) method,
the hydrothermal method can accurately adjust the morphology of the particles.In the
hydrothermal reaction, the formation of cerium oxide particles experienced the
hydrolysis of cerium salt, the nucleation and growth of particles. Nucleation and
dissolution compete with each other, not a simple oxidation reaction. In this paper,
rod-like and cubic ceria react and grow under strong alkali hydrothermal conditions.
Different temperatures lead to different morphologies. In this strong alkali
environment, only ceria with these two morphologies can be obtained without adding
inducers. The octahedron is obtained because sodium phosphate is added as an
inducer, and the particles can grow into an octahedron morphology. The octahedral
structure has less oxygen vacancies and stable structure, so the proportion of trivalent
cerium is the least.]
Comments 2: [In the introduction, the authors should revise the literature review to ensure its
relevance. For example, the statement "Rare earth elements have been applied in the field of
antibacterial materials and achieved good results" is referenced [7]. However, this reference is to a
study on the use of rare-earth–platinum alloy nanoparticles in mesoporous zeolite for the catalytic
propane dehydrogenation reaction.]
Response 2: [Reference [7] has been replaced by other references.]
Comments 3: [The same remark applies to the Results and Discussion section. For instance, the
text "Therefore, the concentration of Ce3+ ions is positively correlated with the increase of lattice
parameters" is accompanied by a reference to research on the green synthesis of CeO2 NPs and
SiO2 NPs from leaf extracts of Nyctanthes arbortristis [24].]
Response 3: [Reference [24] has been replaced by other references.]
Comments 4: [It is also important to note that size-dependent lattice expansion is observed in
many ionic compound nanoparticles. Back in 2012, Diehm et al. showed that surface stress due to
negative Madelung pressure is the main cause of lattice expansion
[https://doi.org/10.1002/cphc.201200257]. They also excluded point defects of various cerium
charge states as a general explanation. One more comment on the mechanisms: On line 402, the
authors state: " CeO2 can bind with sulfhydryl groups on proteins and further intensify the leakage
of cellular contents". Unlike silver, however, cerium has no affinity for sulfhydryl groups. Its
thiophilicity is 0.1, which is too low [see Table S4 in
https://doi.org/10.1021/acs.inorgchem.6b01702]. Future studies should strongly consider using
modern knowledge on the behavior of cerium oxide nanoparticles, including the mechanisms of
antioxidant and antimicrobial activity.]
Response 4: [On the original line 402, the sentence“ CeO2 can bind with sulfhydryl
groups on proteins and the leakage of cellular contents further ”has been deleted.]
Comments 5: [On page 6, line 211, the authors state that the lattice parameter of CeO2-Rod
increases more than those of CeO2-Rod and CeO2-Cube. However, the value cannot exceed itself,
therefore, the typo should be corrected.]
Response 5: [On page 6, line 211, print error has been modified.]
Comments 6: [There are duplicate reference numbers in columns and in parentheses in the
References. The excess numbers need to be removed.]
Response 5: [The excess numbers have been removed.]
Comments 7: [Bacterial species names, such as Escherichia coli, should be italicized in the text.]
Response 7: [All Escherichia coli in the text has been changed to italics.]
Comments 8: [The final sentence of the conclusion reads: "The CeO2 nanostructures reported in
this study… hold promise for diverse applications, including drug delivery, anticancer and
anti-inflammatory therapies, antifungal activity, optoelectronics, gas sensing, and cosmetics".
However, since these claims are not supported by experiments in the submitted manuscript, they
should be removed or relocated to other sections as speculation or discussion.]
Response 8: [This sentence in the conclusion section of the original text has been
deleted “The CeO2 nanostructures reported in this study... hold promise for diverse
applications, including drug delivery, anticancer and anti-inflammatory therapies,
antifungal activity. optoelectronics, gas sensing, and cosmetics”.]
Comments 9: [The text needs to be checked for grammar errors. There are numerous missing
spaces, such as consecutive spellings of words (e.g., line 318: "CeO2-Cubeand"), errors in
superscripts and subscripts, and others. Proofreading by a native speaker is also recommended.]
Response 9: [Grammatical errors have been checked. The missing number of spaces is
also checked and supplemented line by line.]
Reviewer 2 Report
Comments and Suggestions for Authors
The overall article is well written with adequate characterization and relation to catalytic activity. The authors should address the following minor issues:
1) What about the surface area (BET surface area)? After normalizing the surface area, is the effect similar or different between a rod and a cube?
2) Pg 11 line 358 Typing mistake: “oxygen vacancies on (100), (110) and (100) crystal planes are 3.94 eV, 2.62 eV and 4.73’’ should be (100), (110) and (111).
Author Response
Thank you very much for taking the time to review this manuscript. Please
find the detailed responses below and the corresponding
revisions/corrections highlighted/in track changes in the re-submitted files.
Comments 1: [ What about the surface area (BET surface area)? After normalizing the surface
area, is the effect similar or different between a rod and a cube?.]
Response 1: [We have measured the specific surface area of CeO2 nanorods and
nanocubes, which are 77.4 m2 / g and 22.4 m2 / g, respectively. The reason why the
antibacterial ability of cerium oxide nanocubes is weaker than that of nanorods is
mainly due to the low concentration of oxygen vacancies. We have compared
nano-ceria with the same morphology, and adjusted the concentration of oxygen
vacancies by doping. It is found that the antibacterial ability of high oxygen vacancy
concentration is stronger when the specific surface area is the same.]
Comments 2: [ Pg 11 line 358 Typing mistake: “oxygen vacancies on (100), (110) and (100) crystal
planes are 3.94 eV, 2.62 eV and 4.73’’ should be (100), (110) and (111).]
Response 2: [The typing error of Pg 11 line 358 has been modified.]
Reviewer 3 Report
Comments and Suggestions for Authors
- Please change “1×10^5” to “1×105”
- MIC values are strange. For CeO2-Rod the concentration range is 8-32 mg/ml, and MIC is 4 mg/ml. For CeO2-Cube MIC value is 20 mg/ml, but concentration points are 32 and 16 mg/ml.
- Please change “As the PH drops,” to “As the pH drops,”
- Please check enumeration of references because now they are double enumerated, e.g. “1. (1) Fleming,” and so on
- Please check absent spaces throughout the text, e.g. “cm-1are” “460cm” “A460is” “Figure5b” “andCeO2-Oct” “withCeO2-” “and-1.64 eV” and so on
- Please place the reference “Ce3+ [29].” to normal text from superscript
- “More detailed EPR spectra can be found in the supporting information”, supporting information is mentioned but not provied
Author Response
Thank you very much for taking the time to review this manuscript. Please
find the detailed responses below and the corresponding
revisions/corrections highlighted/in track changes in the re-submitted files.
Comments 1: [Please change “1×10^5” to “1×105”]
Response 1: [“1×10^5” has been changed to “1×105”.]
Comments 2: [MIC values are strange. For CeO2-Rod the concentration range is 8-32 mg/ml, and
MIC is 4 mg/ml. For CeO2-Cube MIC value is 20 mg/ml, but concentration points are 32 and 16
mg/ml.]
Response 2: [In the data of Table 1, we selected three representative concentrations for
comparison. When the concentration of octahedral particles was 32 mg / mL, there
was no antibacterial ability. When the concentration of cubic particles was 32 mg /
mL, there was nearly 100 % antibacterial ability, but it decreased to 85 % at 16 mg /
mL, and the antibacterial rate was close to 0 at 8 mg / mL. In contrast, rod-shaped
particles have relatively good antibacterial rate, and there are 100 antibacterial rates at
concentrations of 8,16 and 32 mg / mL. In addition, we also tested and compared the
minimum inhibitory concentration (MIC). The MIC of the rod-shaped particles was 4
mg / mL, and the MIC of the cube was 20mg / mL.]
Comments 3: [Please change “As the PH drops,” to “As the pH drops,”]
Response 3: [“PH” has been changed to “pH”.]
Comments 4: [Please check enumeration of references because now they are double enumerated,
e.g. “1. (1) Fleming,” and so on]
Response 4: [The problems mentioned have been corrected.]
Comments 5: [Please check absent spaces throughout the text, e.g. “cm-1are” “460cm” “A460is”
“Figure5b” “andCeO2-Oct” “withCeO2-” “and-1.64 eV” and so on]
Response 5: [The problems mentioned have been corrected.]
Comments 6: [Please place the reference “Ce3+ [29].” to normal text from superscript]
Response 6: [The problems mentioned have been corrected.]
Comments 7: [“More detailed EPR spectra can be found in the supporting information”, supporting
information is mentioned but not provied]
Response 7: [The sentence “More detailed EPR spectra can be found in the supporting
information” was not necessary and was deleted.]
Round 2
Reviewer 1 Report
Comments and Suggestions for Authors
no additional comments